# Unexpected steric hindrance failure in the gas phase F⁻ + (CH₃)₃CI S$_N$2 reaction

Xiaoxiao Lu[1,3], Chenyao Shang[1,3], Lulu Li[1], Rongjun Chen ⬡[1], Bina Fu ⬡[1] ✉, Xin Xu ⬡[2] & Dong H. Zhang ⬡[1] ✉

Base-induced elimination (E2) and bimolecular nucleophilic substitution (S$_N$2) reactions are of significant importance in physical organic chemistry. The textbook example of the retardation of S$_N$2 reactivity by bulky alkyl substitution is widely accepted based on the static analysis of molecular structure and steric environment. However, the direct dynamical evidence of the steric hindrance of S$_N$2 from experiment or theory remains rare. Here, we report an unprecedented full-dimensional (39-dimensional) machine learning-based potential energy surface for the 15-atom F⁻ + (CH₃)₃CI reaction, facilitating the reliable and efficient reaction dynamics simulations that can reproduce well the experimental outcomes and examine associated atomic-molecular level mechanisms. Moreover, we found surprisingly high "intrinsic" reactivity of S$_N$2 when the E2 pathway is completely blocked, indicating the reaction that intends to proceed via E2 transits to S$_N$2 instead, due to a shared pre-reaction minimum. This finding indicates that the competing factor of E2 but not the steric hindrance determines the small reactivity of S$_N$2 for the F⁻ + (CH₃)₃CI reaction. Our study provides new insight into the dynamical origin that determines the intrinsic reactivity in gas-phase organic chemistry.

Bimolecular nucleophilic substitution (S$_N$2) and base-induced elimination (E2) are the most fundamental reactions which are of crucial importance in preparative organic chemistry[1–4]. Gas-phase studies provide a way out for the identification of organic reaction mechanisms independent of solvation effects, thereby investigating the intrinsic factors that determine the reactivity. The S$_N$2 reaction occurs usually via the back-side attack Walden inversion mechanism, where a nucleophile approaches an alkyl halide from one side of the carbon atom (α-carbon), substitutes a leaving group on the opposite side, leading to the inversion of the tetrahedral carbon center. Although more mechanisms like front-side attack, double inversion, and indirect processes were identified recently in some gas-phase S$_N$2 reactions[5–13], the back-side attack Walden inversion mechanism plays a dominant role in those reactions.

In contrast to S$_N$2, the attacking nucleophile abstracts the hydrogen atom in the β position (β-H), and then elimination occurs in the E2 reaction. In a prototype X⁻ + RY reaction involving the attack of

an alkyl halide by a nucleophilic partner, only the S$_N$2 pathway is possible for the simplest methyl halides[5,6,9–13]. Regarding the increasingly methylated alkyl halides, the E2 pathway arises and the competition between S$_N$2 and E2 has been a hot topic of ongoing interest[14–26]. For example, F⁻ is a strong Lewis base, which engages in a strongly stabilizing interaction with the substrate and can overcome the high distortion energy corresponding to E2, and thus the reaction of F- with halothane favors E2 over S$_N$2[24–26].

It is difficult to differentiate the contributions from S$_N$2 and E2 by straightforward gas-phase experiments[27], because the two pathways produce the same ionic product. Particular experimental efforts were devoted to understanding this competition by indirect mass spectrometric estimations[22,28,29], relying on, for instance, the deuterium kinetic isotope effects and nucleophilic dianions. These investigations provide valuable insight into the overall branching ratios of S$_N$2 versus E2 but in the absence of intrinsic dynamics.

[1]State Key Laboratory of Molecular Reaction Dynamics and Center for Theoretical and Computational Chemistry, Dalian Institute of Chemical Physics, Chinese Academy of Sciences, Zhongshan Road 457, Dalian 116023, China. [2]Department of Chemistry, Fudan University, Shanghai 200433, China. [3]These authors contributed equally: X. Lu and C. Shang. ✉e-mail: bina@dicp.ac.cn; zhangdh@dicp.ac.cn

Recently, crossed beam scattering experiments under single-collision conditions were developed in Wester's group to directly image the differential cross sections (DCSs) for a series of reactions between anions and methyl-substituted alkyl halides of the X⁻ + RY type[21,27,30,31]. These experiments, combined with dynamics theory, provided direct evidence for a variety of reaction mechanisms at the atomic-molecular level, and also the competing $S_N2$ and E2 reaction pathways[21,27,30,31]. Particularly, it was observed that the $S_N2$ reactivity decreases for F⁻ reactions with increasing methyl-substituted alkyl halides, which are F⁻ + $CH_3I$, F⁻ + $C_2H_5I$, F⁻ + $^iC_3H_7I$, and F⁻ + $^tC_4H_9I$[27,30]. In particular, for F⁻ + $^tC_4H_9I$, the measured DCSs imply the exclusive contribution from E2, while the $S_N2$ pathway nearly vanishes[30].

The measured $S_N2$ inhibition can be rationalized intuitively with the steric environment: the totally substituted $(CH_3)_3CI$ is so crowded that the reaction cannot occur because the nucleophile is unable to approach enough the shielded central carbon atom to do a back-side attack[16,32–36]. These experimental findings were attributed simply to the steric hindrance at the $\alpha$-carbon, which is a textbook example of the retardation of $S_N2$ reactivity by bulky alkyl substitution. Nevertheless, the assumed steric effects cannot be directly observed merely by the experiment, and an accurate characterization from the full-dimensional dynamics simulations which can reproduce experimental results and examine the steric effects is highly desirable. A total of 15 atoms are included and 39 degrees of freedom should be considered for the F⁻ + $^tC_4H_9I$ reaction, posing big challenges to construct an accurate, global PES for efficient dynamics simulations. Here we developed an accurate full-dimensional (39-dimensional) PES for this 15-atom F⁻ + $^tC_4H_9I$ reaction and carried out extensive quasiclassical trajectory (QCT) simulations. The dynamical simulations with the aid of this global PES not only reproduce the experimental measurements but also uncover the true dynamical origin of the exceptionally small $S_N2$ reactivity and high E2 reactivity as well as associated mechanisms for the F⁻ + $^tC_4H_9I$ reaction.

## Results

### 39-dimensional potential energy surface

The current 39-dimensional PES of the title reaction was developed by the recently proposed fundamental invariant neural network (FI-NN)[37–41] approach. Due to the huge computational cost of the gold standard coupled-cluster calculations for this 15-atom system with six heavy atoms, we employed the CAM-XYG3/AVTZ(AVTZ-PP for iodine atom)[42,43] method alternatively, which archives a similar level of accuracy as compared to CCSD(T)/AVTZ(-PP) but significantly reduces the computational effort. Comparisons between the CCSD(T)/AVTZ(-PP) and CAM-XYG3/AVTZ(-PP) energies for all stationary points of the E2 and $S_N2$ pathways are made in Fig. 1, which shows good agreement between the two results. A total of ~220,000 CAM-XYG3/AVTZ(-PP)[42,43] energy points were included in the fitting data set. The overall root mean square error (RMSE) of the fitted PES is only 8.3 meV, representing the high accuracy of the current fitting and 15-atom multi-channel PES as well as the strong capability of the FI-NN approach. In addition, we employed the space partitioning and energy splitting methods to overcome the huge challenge to construct a 39-dimensional multi-channel reaction with long-range interaction in the asymptotes. All details of the fitting approach and properties of the PES are given in the Supplementary Information. The distribution of fitting errors and optimized geometries are shown in Supplementary Figs. 1, 2, respectively.

The F⁻ + $(CH_3)_3CI$ reaction involves the two following important product channels:

$$F^- + (CH_3)_3 CI \rightarrow (CH_3)_2CCH_2 + HF + I^- \quad (E2)$$
$$F^- + (CH_3)_3 CI \rightarrow (CH_3)_3 CF + I^- \quad (S_N2) \quad (1)$$

As depicted in Fig. 1, we can find that both E2 and back-side attack $S_N2$ reactions are barrierless and highly exothermic, associated with pre-reaction wells on the reactant side and post-reaction wells on the product side. There is a high barrier (TS0) pathway, which

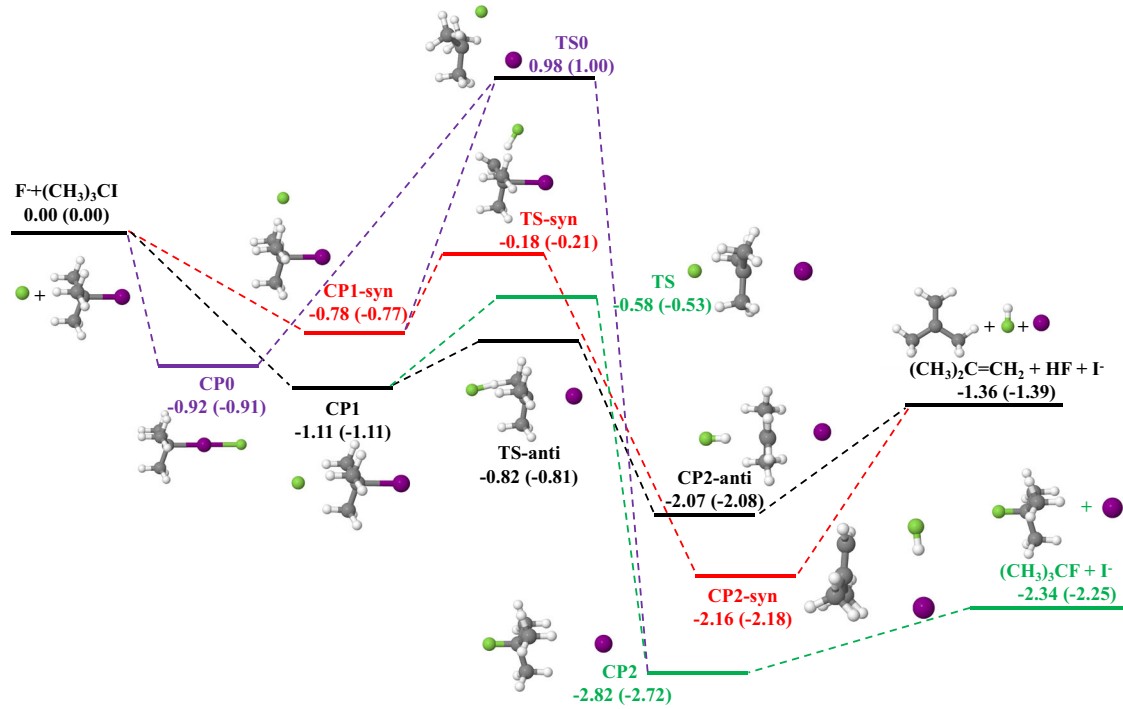

**Fig. 1 | Schematic PES of the F⁻ + $(CH_3)_3CI$ reaction.** Green curve: back-side attack $S_N2$; Purple: front-side attack $S_N2$; Black: *anti*-E2; Red: *syn*-E2. The relative energies obtained from CAM-XYG3/AVTZ(-PP) and CCSD(T)/AVTZ(-PP) (in brackets) methods are all in eV.

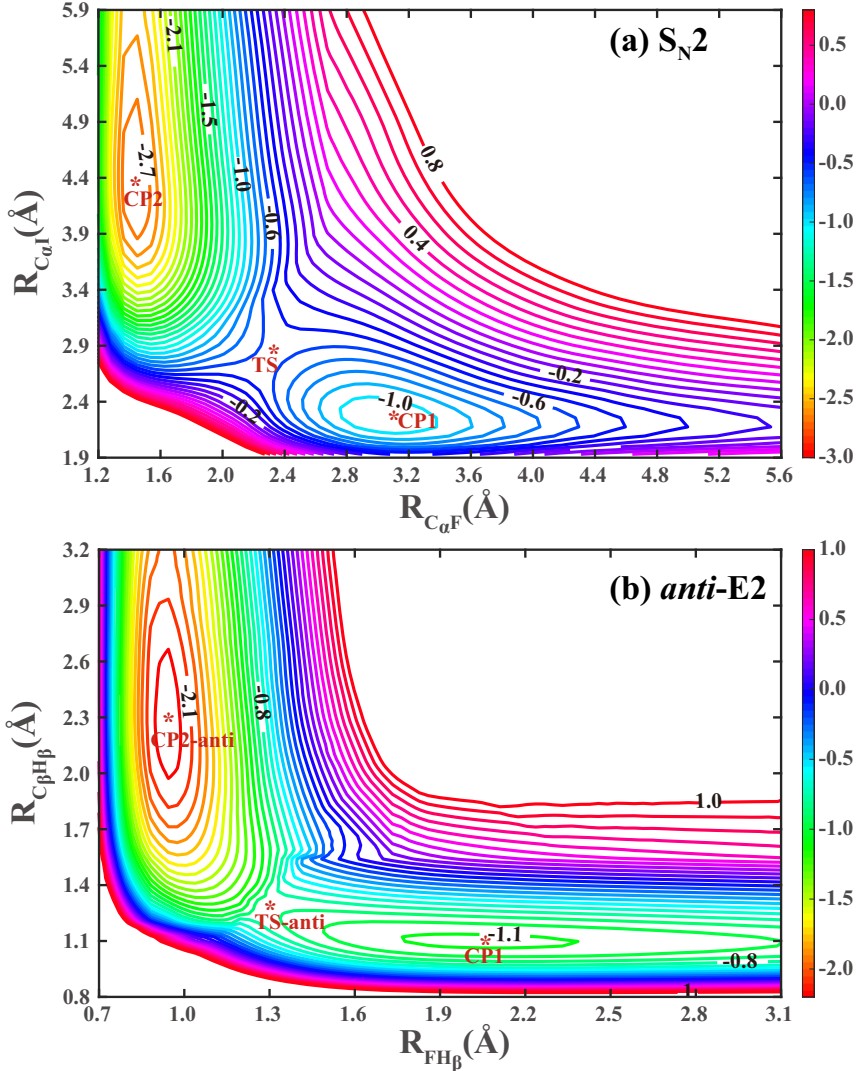

**Fig. 2 | Contour plots.** The contour plots of back-side attack $S_N2$ (**a**) and *anti*-E2 (**b**) channels on the FI-NN PES by full-dimensional optimization.

corresponds to the front-side attack $S_N2$, with a slightly shallower pre-reaction minimum (CP0) than CP1. The E2 reaction undergoes either a lower energy pathway of *anti*-E2 or a higher one of *syn*-E2, indicating the *anti*-E2 pathway is more kinetically favorable. We note that the $S_N2$ reaction shares an ion-dipole bound pre-reaction complex (CP1) with the *anti*-E2 pathway, which lies −1.11 eV below the reactant asymptote. The submerged barrier for $S_N2$ is 0.24 eV higher than that for *anti*-E2, but the exothermic energy for $S_N2$ is much larger than that for E2.

To further demonstrate the accurate behavior of full-dimensional PES, we show two contour plots of back-side attack $S_N2$ and *anti*-E2 pathways in Fig. 2. The contours of front-side attack $S_N2$ and *syn*-E2 pathways with higher energy profiles are depicted in Supplementary Figs 3, 4, respectively. These contours are plotted using the full-dimensional optimization on the global full-dimensional PES. As seen, these contours are quite smooth, and the position and energies of reactants, pre-reaction minimums, transition states, post-reaction minimums, and products, are all well described. To the best of our knowledge, this is also the first time these contours can be made for such large reactive systems. These contours further reflect the high accuracy of the current fitting and reliability of 15-atom multi-channel PES, as well as the strong capability of the FI-NN approach. We can go much further than predicting reaction pathways based on stationary points with an accurate, full-dimensional, analytical PES, and produce observable outcomes and ultimately provide deep insight into the dynamical mechanisms of competing $S_N2$ and E2 pathways and associated steric effects for this ion-molecule reaction.

### Reaction dynamics simulations
Standard QCT calculations were carried out at the collision energies ($E_c$) ranging from 0.2 to 1.9 eV based on the full-dimensional PES. A total of 4.2 million trajectories were run for $(CH_3)_3CI$ initially in the ground rovibrational state for each collision energy. It is worth mentioning that we developed the computational approach to get analytical gradients from the FI-NN PES, which raises the computational speed by about ten times as compared to the numerical gradients, and facilitates the calculations of a large number of trajectories for good statistics. Detailed information on QCT calculations are given in the Supplementary Information.

We found the collision of $F^-$ with $(CH_3)_3CI$ leads to nearly the E2 product channel, while the cross section of $S_N2$ is roughly 50 times smaller than that of E2 at all the collision energies, indicating the $S_N2$ reactivity can be negligible in resulting in $I^-$ product ions. In addition, the limited $S_N2$ reactivity results from exclusively the back-side attack mechanism, and no front-side attack mechanism was seen, presumably due to the much higher barrier. Thus, "$S_N2$" below refers to back-side attack $S_N2$ for simplicity. This finding is consistent with the recent experiment, which indicates the exclusive E2 reaction[30].

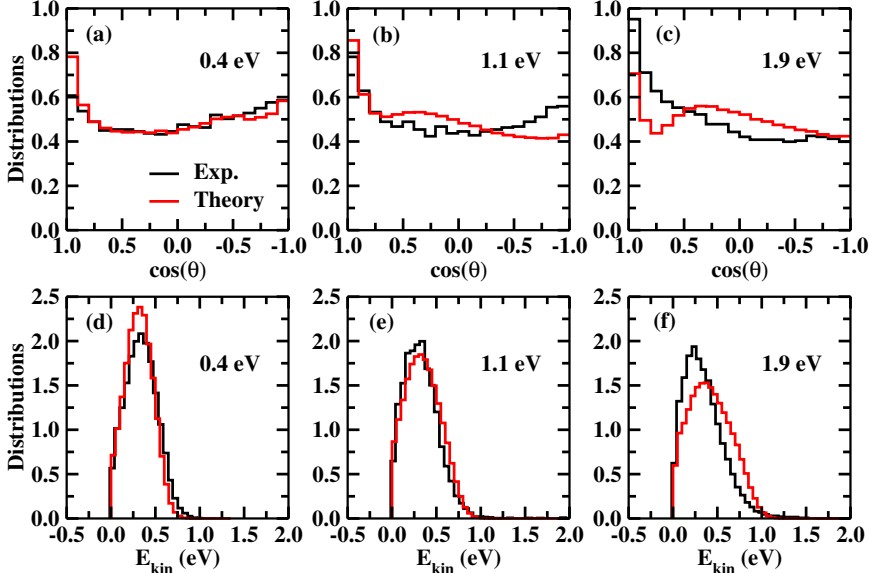

**Fig. 3 | Theory and experiment.** Comparisons of normalized angular distributions (**a**–**c**) and translational energy distributions (**d**–**f**) of product I⁻ in the center of the mass frame at the collision energy of 0.4, 1.1, and 1.9 eV between the experimental measurement and QCT calculations.

## Product angular and translational energy distributions

A comparison of detailed dynamics from theory (PES and dynamics) and experiment can test the reliability of the former. Figure 3 shows the computed center-of-mass (CM) angular and translational energy distributions for I⁻ product ions, together with the experimental results[30] at three collision energies. We found a very good agreement between theory and experiment.

At the lowest collision energy of 0.4 eV, both theoretical and experimental angular distributions are almost forward-backward symmetric, with slightly more intensity in the forward direction from theory. This is associated with the dominant indirect E2 mechanism at low collision energies, due to the pre-reaction minimum. The intensity in the forward direction from the QCT calculations results from the contributions from a small fraction (-11%) of the direct E2 mechanism. With the increasing collision energy, the computed angular distributions exhibit more intensity in the forward and sideways directions, consistent with the experimental results showing more signals in the forward scattering. We found the direct mechanism of E2 increases when the collision energy increases in the QCT simulations, as shown in Supplementary Fig. 5 for the cross sections of direct and indirect pathways, and this trend was also predicted by the experiment[30]. Due to the lower minimum energy path of *anti*-E2, the reaction via the *anti*-E2 pathway dominates over *syn*-E2 (Supplementary Fig. 6). Animations of typical indirect and direct trajectories via *anti*-E2 and *syn*-E2 are given in the supplementary Movies 1–4.

The agreement between theory and experiment for the translational energy distributions of I⁻ is impressive (Fig. 3d–f). Overall, the three translational energy distributions show nearly invariant with respect to the collision energies, which all peaks at around 0.3 eV and vanishes at around 1.0 eV. The product translational energy release is independent of the reactant collision energy, supporting that the complex-forming indirect dynamical process leads to intramolecular rovibrational energy redistribution among different modes of the complex. Therefore, the E2 products are rovibrationally excited, and most available energies are channeled into the rovibrational modes of products. The internal energy distributions of the product $(CH_3)_2CCH_2$ are shown in Supplementary Fig. 7. The distribution peaks at around 3.5–4.0 eV, showing a tail up to 5.0–6.0 eV. The zero-point energy of $((CH_3)_2CCH_2$ is only 1.9 eV. Thus, the $(CH_3)_2CCH_2$ product is highly rovibrationally excited. In addition, the $(CH_3)_2CCH_2$ product becomes more rovibrationally excited as the increase of collision energy.

## S$_N$2 versus E2

Since the theory is capable of reproducing the angular distributions and translational energy distributions of I⁻, now we are confident to provide deep insight into the dynamical origin of the quite small reactivity of the S$_N$2 reaction. Although the steric environment of $\alpha$-carbon in the F⁻ + $^tC_4H_9I$ reaction seems crowded, the attacking F⁻ is relatively small. In addition, F⁻ has strong nucleophilicity (F⁻ > Cl⁻ > Br⁻ > I⁻) and I⁻ is a good leaving group (the energy of the C-I bond is only 50% of the C-H bond). As discussed above, the S$_N$2 reaction is connected to E2 with a shared pre-reaction minimum, followed by a slightly higher submerged barrier than that for *anti*-E2, but a more deep well of post-reaction minimum and lower product energies. Therefore, the near vanish of S$_N$2 is unexpected, and the assumption of complete steric hindrance of the S$_N$2 reactivity remains in doubt.

## "Intrinsic" reactivity of S$_N$2

It is important to find out the "intrinsic" reactivity of S$_N$2 regardless of the competing E2, which can be accomplished based on the current PES. We propose a numerically simple but efficient way to block the E2 pathway but nothing is changed for S$_N$2, by adding a repulsive potential between F⁻ and $\beta$-H in the vicinity of the E2 transition state of the global PES. The form of the repulsive potential is expressed as follows.

$$\Delta V = \frac{1}{10 \times (1 + \exp(10 \times (R_{FH} - 1.3)))}. \tag{2}$$

Here, $R_{FH}$ is the distance between F⁻ and $\beta$-H. Figure 4 shows the minimum energy paths of the S$_N$2 reaction obtained from the original FI-NN PES and the modified PES, which shows excellent agreement between them. The minimum energy paths were determined by the quadratic steepest descent method[44]. The two minimum energy paths were also verified by the direct CAM-XYG3/AVTZ(-PP) calculations. We have also shown the contour plot of the S$_N$2 reaction on the modified PES in Supplementary Fig. 8, which is exactly the same as the contour plot on the original full-dimensional PES (Fig. 2a). In addition, we have randomly selected two S$_N$2 trajectories and calculated the potential energies of configurations along the two trajectories. As shown in

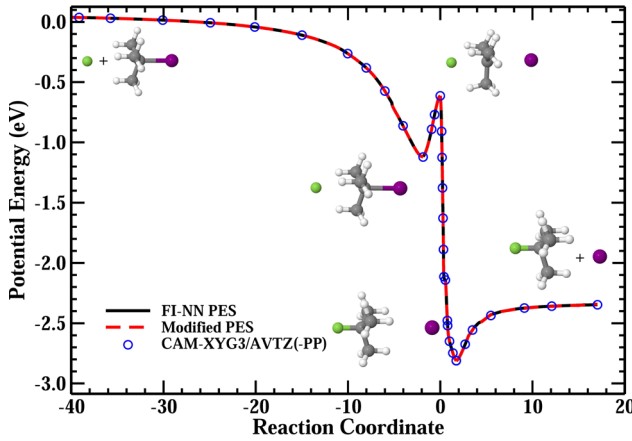

**Fig. 4 | Minimum energy paths.** The minimum energy paths of the $S_N2$ channel on the FI-NN PES and the modified PES, together with the corresponding CAM-XYG3/AVTZ(-PP) energies.

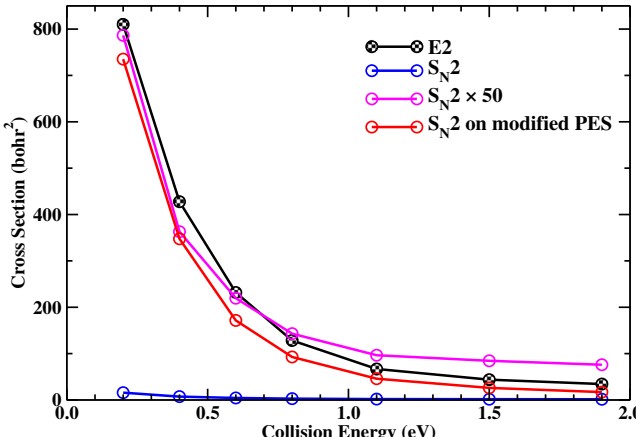

**Fig. 5 | Integral cross sections.** Cross sections as a function of collision energy for the E2 and $S_N2$ reactions on the FI-NN PES and those for the $S_N2$ reaction on the modified PES. A repulsive potential between $F^-$ and $\beta$-H in the vicinity of the E2 transition state of the FI-NN PES was added in the modified PES, which blocks the E2 pathway, but nothing is changed for $S_N2$ as presented above.

Supplementary Fig. 9, the energies from the original FI-NN and modified PESs are basically the same, and they are all well reproduced by CAM-XYG3/AVTZ(-PP) calculations. The E2 scattering events were no longer found in QCT calculations on the modified PES, indicating the E2 channel is completely blocked. As a result, the impose of Eq. (2) on the FI-NN PES leads to a modified PES, which does not change the full-dimensional PES of $S_N2$ or the behavior of all 39 degrees of freedom of $S_N2$, but completely blocks the E2 pathway.

## Cross sections of $S_N2$ and E2

The computed integral cross sections for the E2 and $S_N2$ pathways on the FI-NN PES as well as on the modified PES are shown in Fig. 5. The $S_N2$ reactivity is roughly 50 times smaller than the E2 reactivity on the FI-NN PES, but the situation changes dramatically on the modified PES. It is surprising that the $S_N2$ reactivity rises to almost the same magnitude as the original E2 reaction, after the E2 channel was completely blocked on the modified PES, which reflects that those trajectories that intended to proceed via the E2 pathway now goes to the $S_N2$ pathway instead, due to a shared pre-reaction minimum. This fact is in contradiction with the assumption that the steric hindrance suppresses the $S_N2$ reactivity since the steric effect on $\alpha$-C associated with $S_N2$ remains unchanged after modifying the PES. Further analysis shows the $S_N2$ reaction proceeds via the direct or indirect back-side attack mechanism. The direct mechanism rises with the increase of collision energy. It is evident that the rather small reactivity of $S_N2$ for the title reaction is due to the inevitable competition with the highly reactive E2 pathway, but not the steric hindrance. Animations of typical indirect and direct $S_N2$ trajectories are given in Supplementary Movies 5, 6.

It is also important to compare the "intrinsic" $S_N2$ reactivity of $F^-$ + $(CH_3)_3CI$ with that of $F^-$ + $CH_3I$. Supplementary Fig. 10 shows the cross sections of $F^-$ + $(CH_3)_3CI$ on the modified PES and those of $F^-$ + $CH_3I$ on the PES developed by ref. 5 as well as on another PES developed by us. The cross sections of $F^-$ + $CH_3I$ on two different PESs are consistent, indicating both the two PESs are sufficiently accurate. It is interesting that the "intrinsic" $S_N2$ reactivity of $(CH_3)_3CI$ is even larger than that of $CH_3I$, probably due to the more exothermic energy of $(CH_3)_3CI$. These results unambiguously show the $S_N2$ reactivity doesn't decrease as the substrate becomes bulkier, which is also evidence of steric hindrance failure.

## PES and dynamics for $Cl^-$ + $(CH_3)_3CI$

In addition to the title $F^-$ + $(CH_3)_3CI$ reaction, we investigated one more reaction, $Cl^-$ + $(CH_3)_3CI$, to make our claim stronger.

Supplementary Fig. 11 shows the schematic of the PES, with important stationary-point geometries and energies illustrated. We can see a different minimum energy path of $Cl^-$ + $(CH_3)_3CI$, as compared with that of $F^-$ + $(CH_3)_3CI$. For instance, the barriers are much higher, the pre-reaction and post-reaction wells are shallower, and the products are not that exothermic for the $Cl^-$ involved E2 and $S_N2$ reactions. Specifically, the $S_N2$ pathway has an overall barrier of 0.23 eV, which is slightly higher than the barrier of 0.17 eV for the E2 pathway. The $S_N2$ reaction shares an ion-dipole bound pre-reaction well with the E2 pathway, which lies −0.74 eV below the reactant asymptote. The $S_N2$ reaction is exothermic, but the E2 pathway is endothermic, with the product energy 0.31 eV above the reactant energy.

The new PES was also modified by adding a similar repulsive potential as has been used for the $F^-$ reaction, which blocks the E2 pathway completely, but nothing is changed for $S_N2$. Standard QCT calculations for $Cl^-$ + $(CH_3)_3CI$ were carried out on the new PES and also on the modified PES at collision energies ranging from 0.4 to 1.1 eV, for the $(CH_3)_3CI$ reactant initially in the ground rovibrational state. A total of roughly 1.3 million trajectories were run. The computed integral cross sections for the E2 and $S_N2$ pathways on the new PES as well as on the modified PES are shown in Supplementary Fig. 12. The $S_N2$ cross section on the new PES is only four times smaller than that of E2, indicating the reactivities of the two pathways are in the same order of magnitude. This is understandable because the $S_N2$ pathway has a higher barrier, although it is more exothermic. After the E2 pathway was completely blocked on the modified PES, the $S_N$ reactivity rises remarkably by about a factor of 2, which is about one-half of the E2 reactivity. Hence the "intrinsic" $S_N2$ reactivity is closely related to the competition of E2 for $Cl^-$ + $(CH_3)_3CI$, but not the steric hindrance, as we concluded for $F^-$ + $(CH_3)_3CI$.

## Discussion

Regarding the high reactivity of the E2 pathway, there are 9 hydrogen atoms in three methyl groups (here the bulk for the title reaction), either of which can react with the attacking $F^-$. If the bulk consists of non-reactive groups, just as we want to demonstrate a similar way in this work by blocking the E2 pathway but maintaining the existence of bulk, the incoming $F^-$ can attack the $\alpha$-carbon easily, and the $S_N2$ reactivity can reach very high. As a result, the nearly vanish of the $S_N2$ reactivity is not due to the steric congestion of the bulk, but the high reactivity of the bulk, which consumes all incoming $F^-$. In other words,

if the E2 reactivity was not that high to consume all incoming $F^-$, one would observe the SN2 reactivity comparable to E2.

To summarize, we have unraveled the dynamical origin of the very small $S_N2$ reactivity in the gas-phase $F^- + {}^tC_4H_9I$ reaction, by developing an accurate 39-dimensional PES and performing QCT simulations based on this global PES. The experimental angular and translational energy distributions are all well reproduced. To the best of our knowledge, this is the first time the dynamics simulations based on a reliable full-dimensional PES can yield accurate dynamics information for a reactive system with up to 15 atoms. As opposed to the textbook example of the steric retardation of $S_N2$ reactivity by bulky substitution, it was found that the suppression of $S_N2$ for the prototype $F^- + {}^tC_4H_9I$ reaction results from the high reactivity of the three methyl groups, i.e., E2 pathway. With the aid of current analytical PES, we verify that in the absence of E2 competition, the "intrinsic" reactivity of $S_N2$ can reach that high even though there exists steric congestion. Our work might provide a possible way to raise the $S_N2$ reactivity by suppressing the reactivity of the competing pathway rather than changing the steric environment of $S_N2$ in physical organic chemistry.

For the $F^- + (CH_3)_3CI$ gas-phase reaction, the transition states are much lower than reactants, i.e. submerged barrier. The reason why the reactions almost proceed to $S_N2$ when E2 is blocked is because the pre-reaction complex well is deep, thus crossing the $S_N2$ transition state to form $S_N2$ products is much easier than dissociating back to reactants, especially at low collision energies. For the gas-phase $Cl^- + (CH_3)_3CI$, the transition state of $S_N2$ is about 0.23 eV higher than reactants and the pre-reaction complex well is not as deep, hence some fractions of reaction would dissociate back to reactants and some would proceed to $S_N2$ when E2 is blocked. We expect a similar scenario in other gas-phase $S_N2$/E2 competitive reactions, whose transition states are much higher than reactants.

## Methods

### Potential energy surface

The full-dimensional PES for the $F^- + {}^tC_4H_9I$ reaction was developed by the FI-NN fitting approach based on a total of roughly 220,000 CAM-XYG3/AVTZ(-pp) energy points. Because the configuration space of the investigated system is very large, we employed the space partitioning and energy splitting methods to overcome the huge difficulties of fitting all the data points. The overall fitting error is only 8.3 meV, measured in terms of root mean square error, indicating the PES is highly accurate. The theoretical and computational approaches used for $Cl^- + (CH_3)_3CI$ are basically the same as those for $F^- + (CH_3)_3CI$. A full-dimensional PES of $Cl^- + (CH_3)_3CI$ was developed by FI-NN fitting to about 270 000 data points. The overall fitting error of the fitted $Cl^- + (CH_3)_3CI$ PES is 18.7 meV. In addition, a new computational approach was developed to produce the analytical gradients of the FI-NN PES, which facilitates efficient and accurate dynamics simulations.

### QCT calculations

Standard QCT calculations for the $F^- + (CH_3)_3CI$ reaction were carried out at collision energies ranging from 0.2 to 1.9 eV on the FI-NN PES, for the $(CH_3)_3CI$ reactant initially in the ground rovibrational state. We randomly sampled the normal coordinates and momenta to get the initial coordinates and momenta of $(CH_3)_3CI$. Adjustments were then made to the momenta to force the angular momentum of $(CH_3)_3CI$ to zero. All trajectories were run using the Velocity-Verlet integration algorithm with a time step of 0.024 fs for a maximum time of 25 ps. We terminated the trajectory when any two fragments reach a separation of 30 Bohr, ending up with the formation of $(CH_3)_2C=CH_2 + HF + I^-$ or $(CH_3)_3CF + I^-$ or returning to the reactants. A total of roughly 4.2 million trajectories were run to obtain the detailed dynamical information of the product $I^-$ at each collision energy.

## Data availability

The data presented in the analysis can be reproduced using code which is available from the corresponding author on reasonable request.

## Code availability

The code used for the analysis in the current study is available from the corresponding author on reasonable request.

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

## Acknowledgements

This work was supported by the National Natural Science Foundation of China (Grant Nos. 22288201, 22173099), LiaoNing Revitalization Talents Program (XLYC1907190).

## Author contributions

D.H.Z. and B.F. conceived and supervised the research; X.L., L.L., C.S., and R.C. performed the research; X.L., L.L., B.F., X.X., and D.H.Z. analyzed the data; and B.F. and D.H.Z. wrote the manuscript.

## Competing interests

The authors declare no competing interests.
