## [Peer Review File · Nature Communications]

Reviewers' Comments:

Reviewer #1:

Remarks to the Author:

The authors report a quasi-classical trajectory study on the dynamics of the competing SN2 and E2 channels of the $F^- + (CH_3)_3CI$ reaction using a newly-developed full-dimensional potential energy surface (PES). With a clever idea, the authors show that SN2 is suppressed by E2 and not by steric hindrance. The paper is well-written, the accurate 39-dimensional PES is unprecedented, the results are interesting, impressive, and unexpected, as well as provide new insights into the dynamics of SN2/E2 systems, which may capture the attention of the broad readership of Nature Communications. Thus, I strongly recommend this work for publication, once the authors address the mostly minor comments given below.

"Base-induced elimination" should be "base-induced elimination" on page 2.

In Figure 1 "0.0" should be "0.00".

In the entrance channel there should be a halogen-bonded $F \cdots IC(CH_3)_3$ complex, whose energy may be deeper than CP1. This halogen-bonded front-side complex may have significant effects on the dynamics of the title reaction, see J. Phys. Chem. Lett. 8, 2917-2923 (2017). How well is this minimum described on the present PES?

In Figures 4, S3, and S4 (and also in the Methods section and in the SI) "Bohr" should be "bohr". In the SI there is a "Bohr" typo as well.

In the caption of Figure 4 the meaning of the modified PES should be briefly defined to help the readers and to increase the significance of the results.

At Ref. 7 "1–6" should be replaced with "5972".

At Ref. 28 "1–5" should be replaced with "13, 977-981".

The titles at the references are written with mixed style (title case and sentence case). This should be consistent following the journal style, which is sentence case to the best of my knowledge.

In the SI at "explicitly correlated coupled-cluster (CCSD(T)) method" CCSD(T)-F12 should be written.

CCSD(T)/AVTZ(-PP) or CCSD(T)-F12/AVTZ(-PP)?

In Figure S2 it would be useful to show the distances at the CAM-XYG3/AVTZ(-PP) level next to the PES values like in the case of the relative energies.

"6-31+g*" should be "6-31+G*"

Ref. 2 of the SI is incomplete.

In Ref. 5 of the SI "???" should be updated.

Reviewer #2:

Remarks to the Author:

Zhang and co-workers studied, in their contribution to Nature Communications, the SN2 reaction between F⁻ and (CH₃)Cl by developing a full-dimensional PES for this reaction and carried out extensive QCT simulations. Although this work has been carried out with care, the conclusions obtained are not physically sound. The authors state that the poor reactivity of the studied SN2 reaction is not due to the increased steric congestion around the alpha carbon but due to the competing E2 reaction.

Various studies have shown that F⁻ prefers, even for the reaction between F⁻ and haloethane, the E2 reaction over the SN2 reaction [J. Am. Chem. Soc. 1991, 113, 6041; J. Chem. Theory Comput. 2009, 5, 1597; Chem. Eur. J. 2020, 26, 15538]. As mentioned by these authors, this is so because F⁻ is a strong Lewis base which engages in a strongly stabilizing interaction with the substrate and, thus, is able to overcome the intrinsically high distortion energy corresponding to the E2 reaction; hence, F⁻

favours E2 over SN2. In other words, F⁻ already intrinsically prefers E2, independent of the steric bulk around the alpha carbon.

Increasing the steric bulk around alpha carbon will suppress, due to increasing steric repulsion between the steric bulk and the nucleophile, the SN2 pathway and thus raise the SN2 barrier [J. Org. Chem. 2007, 72, 5660; J. Am. Chem. Soc. 2010, 132, 3808; J. Am. Chem. Soc. 2009, 131, 16162; J. Am. Chem. Soc. 2004, 126, 9054.]. The E2 pathway, on the other hand, does not get affected much by increasing the steric bulk around the alpha-carbon and thus, the E2 barrier increases only slightly. This will lead to a larger difference in barrier height between the E2 and SN2 reaction pathways and hence favours the E2 even more, as seen in this work. Thus, the steric bulk around the carbon does, in fact, play a role in the low reactivity of the SN2 pathway.

Interestingly, the authors state in their Discussion section that the steric bulk makes the SN2 reaction more difficult to occur. This is in line with the current textbook explanation behind the retardation of the SN2 reactivity by bulky alkyl substituents and opposes the main conclusion of the authors.

Due to the above-mentioned inconsistencies in this work, I cannot recommend the publication of this manuscript in Nature Communications.

Reviewer #3:

Remarks to the Author:

In this article, the authors report an unprecedented full-dimensional (39-dimensional) machine learning-based potential energy surface for the 15-atom F⁻ + (CH₃)₃I reaction, and the QCT simulation is carried out to study the elimination (E2) and bimolecular nucleophilic substitution (SN2). I have the following comments for this article.

1. This is a multi-dimensional potential (39-dimensional), but the authors only show a schematic plot of the PES, without a contour plot of the important coordinates of the reaction, one could not see the relative positions of transition states (TS). In the schematic PES, all the TS are in the same position, which is not possible. Maybe the findings of the study also depend on the positions of the TS. Investigation should be done on this.

2. The authors state a full dimensional machine-learning PES, why does this reaction only have E2 and SN2 mechanisms on the full-d PES? I am not convinced there are only two mechanisms presented for this system. The front-side attack? the double-inversion? Other mechanisms might occur but have

the same product as the two mechanisms, which leads to the suppression of one and the enhancement of the other of the two mechanisms here.

3. The authors also add a repulsive potential between F-H in the vicinity of the E2 transition state of the global PES. The Equ (2) is a one-dimensional, however the reaction is 39-dimensional with all the coordinates coupled, how do the authors add this potential without changing the behavior of other coordinates?

Without a thorough study of the possible mechanisms on the PES, I am not convinced the authors' argument is strong enough to support the current conclusion.

We thank the referees for taking the time to provide constructive comments to our manuscript. The detailed revisions to the manuscript are indicated by red fonts, and the point-by-point response to the comments are presented as follows. All the referees' comments are reproduced before the responses (indicated by blue fonts).

Reviewers' comments:

Reviewer #1 (Remarks to the Author):

The authors report a quasi-classical trajectory study on the dynamics of the competing SN2 and E2 channels of the F⁻ + (CH₃)₃CI reaction using a newly-developed full-dimensional potential energy surface (PES). With a clever idea, the authors show that SN2 is suppressed by E2 and not by steric hindrance. The paper is well-written, the accurate 39-dimensional PES is unprecedented, the results are interesting, impressive, and unexpected, as well as provide new insights into the dynamics of SN2/E2 systems, which may capture the attention of the broad readership of Nature Communications. Thus, I strongly recommend this work for publication, once the authors address the mostly minor comments given below.

"Base-induced elimination" should be "base-induced elimination" on page 2.

Fixed!

In Figure 1 "0.0" should be "0.00".

Fixed!

In the entrance channel there should be a halogen-bonded F...IC(CH₃)₃ complex, whose energy may be deeper than CP1. This halogen-bonded front-side complex may have significant effects on the dynamics of the title reaction, see J. Phys. Chem. Lett. 8, 2917-2923 (2017). How well is this minimum described on the present PES?

We have added the halogen-bonded F...IC(CH₃)₃ complex in Fig. 1 as CP0. We found CP0 is about 0.92 eV below the reactant, which is shallower than CP1. This CP0 complex is connected with the front-side attack transition state of S_N2, as TS0 in Fig. 1. The TS0 lies high above the reactant, with a barrier height of about 1.0 eV.

In Figures 4, S3, and S4 (and also in the Methods section and in the SI) "Bohr" should be "bohr". In the SI there is a "Bohr" typo as well.

Fixed!

In the caption of Figure 4 the meaning of the modified PES should be briefly defined to help the readers and to increase the significance of the results.

We have added the following sentence in the caption of Fig. 4. "A repulsive potential between F⁻ and β-H in the vicinity of the E2 transition state of the FI-NN PES was

added in the modified PES, which blocks the E2 pathway but nothing is changed for S_N2 as presented above.”

At Ref. 7 "1–6" should be replaced with "5972".

Fixed!

At Ref. 28 "1–5" should be replaced with "13, 977-981".

Fixed!

The titles at the references are written with mixed style (title case and sentence case). This should be consistent following the journal style, which is sentence case to the best of my knowledge.

We have changed the style of all the references to be in the consistent sentence case.

In the SI at "explicitly correlated coupled-cluster (CCSD(T)) method" CCSD(T)-F12 should be written.

Fixed!

CCSD(T)/AVTZ(-PP) or CCSD(T)-F12/AVTZ(-PP)?

Here we used CCSD(T)/AVTZ(-PP).

In Figure S2 it would be useful to show the distances at the CAM-XYG3/AVTZ(-PP) level next to the PES values like in the case of the relative energies.

We have added the distances optimized at the XYGJ-OS/AVDZ(-PP) level. It is extremely time consuming to directly perform the optimization using CAM-XYG3/AVTZ(-PP), because numerical forces have to be calculated due to the unavailability of analytical forces for the method. The energies are calculated at the CAM-XYG3/AVTZ(-PP) level. The geometries optimized by XYGJ-OS /AVDZ(-PP) should be very similar to those by CAM-XYG3/AVTZ(-PP). We can see good agreement between the XYGJ-OS /AVDZ(-PP) calculations and the PES for the geometries.

"6-31+g*" should be "6-31+G*"

Fixed!

Ref. 2 of the SI is incomplete.

Fixed!

In Ref. 5 of the SI "???" should be updated.

Fixed!

Reviewer #2 (Remarks to the Author):

Zhang and co-workers studied, in their contribution to Nature Communications, the S_N2 reaction between F^- and $(CH_3)Cl$ by developing a full-dimensional PES for this reaction and carried out extensive QCT simulations. Although this work has been

carried out with care, the conclusions obtained are not physically sound. The authors state that the poor reactivity of the studied SN2 reaction is not due to the increased steric congestion around the alpha carbon but due to the competing E2 reaction.

Various studies have shown that F⁻ prefers, even for the reaction between F⁻ and haloethane, the E2 reaction over the SN2 reaction [J. Am. Chem. Soc. 1991, 113, 6041; J. Chem. Theory Comput. 2009, 5, 1597; Chem. Eur. J. 2020, 26, 15538]. As mentioned by these authors, this is so because F⁻ is a strong Lewis base which engages in a strongly stabilizing interaction with the substrate and, thus, is able to overcome the intrinsically high distortion energy corresponding to the E2 reaction; hence, F⁻ favors E2 over SN2. In other words, F⁻ already intrinsically prefers E2, independent of the steric bulk around the alpha carbon.

We completely agree with the referee that the E2 reaction is preferred for many reactions, including the current F⁻+(CH₃)₃CI reaction. We thank the referee for listing several papers about the X⁻+C₂H₅Y reaction, and we have included these papers as refs. [24-26] and cited them on Page 2 after the following sentence. “Regarding the increasingly methylated alkyl halides, the E2 pathway arises and the competition between S_N2 and E2 has been a hot topic of ongoing interest [14–26]. For example, F⁻ is a strong Lewis base which engages in a strongly stabilizing interaction with the substrate and can overcome the high distortion energy corresponding to E2, and thus the reaction of F⁻ with haloethane favors E2 over S_N2[24-26].”

We note that these literatures are related to stationary point calculations along the reaction paths. These static investigations, including the energies of barriers, intermediates and products, as well as orbital interaction and activation strain analysis, provide important insight into the competition between E2 and S_N2.

Now for the much larger F⁻+(CH₃)₃CI reaction, we may conjecture it also favors E2 over S_N2 based on previous literatures. However, it is extremely interesting that the recent ion-imaging experiment by Roland Wester found the S_N2 pathway nearly vanishes at a wide range of collision energies for the F⁻+(CH₃)₃CI reaction (JACS, 2019, 141, 20300). We did describe the details on Page 2 and Page 3. As also seen from the energetics in Fig. 1 of our manuscript, the S_N2 reactivity can be smaller than anti-E2, but should not be neglected.

Is the varnish of S_N2 due to the steric hindrance at the α-carbon? This question can be only resolved by full-dimensional dynamics simulations based on a full-dimensional (39-dimension) PES. We have also elucidated the reaction mechanisms at the atomic-level for the F⁻+(CH₃)₃CI reaction, based on the full-dimensional PES. This is one significant point we want to address in the current manuscript.

Increasing the steric bulk around alpha carbon will suppress, due to increasing steric repulsion between the steric bulk and the nucleophile, the SN2 pathway and thus raise the SN2 barrier [J. Org. Chem. 2007, 72, 5660; J. Am. Chem. Soc. 2010, 132, 3808; J. Am. Chem. Soc. 2009, 131, 16162; J. Am. Chem. Soc. 2004, 126, 9054.]. The E2

pathway, on the other hand, does not get affected much by increasing the steric bulk around the alpha-carbon and thus, the E2 barrier increases only slightly. This will lead to a larger difference in barrier height between the E2 and SN2 reaction pathways and hence favors the E2 even more, as seen in this work. Thus, the steric bulk around the carbon does, in fact, play a role in the low reactivity of the SN2 pathway.

We thank the referee for listing more papers. The papers [J. Am. Chem. Soc. 2009, 131, 16162; J. Am. Chem. Soc. 2004, 126, 9054] were included in the original manuscript, now we have cited another two papers [J. Org. Chem. 2007, 72, 5660; J. Am. Chem. Soc. 2010, 132, 3808] in the revised manuscript as refs. 35 and 36.

We agree with the referee that for those reactions discussed in the literatures, increasing the steric bulk around α carbon can raise the S_N2 barrier, but the E2 barrier can be slightly increased. However, the true reactivity of E2 or S_N2 can only be verified by dynamics calculations.

With the aid of full-dimensional analytical PES, we can add a repulsive potential and block the E2 pathway completely, without changing the PES and steric effect on α -carbon of the S_N2 channel. If the steric congestion plays a role, the S_N2 reactivity should still be small when E2 is blocked. However, the result is totally different! It is surprising that the S_N2 reactivity rises to almost the same magnitude as the original E2 reaction when the E2 channel was completely blocked on the modified full-dimensional PES, which reflects that those trajectories that intended to proceed via the E2 pathway now goes to the S_N2 pathway instead, due to a shared pre-reaction minimum. This fact is in contradiction with the assumption that the steric hindrance suppresses the S_N2 reactivity of this reaction, since the steric effect on α -carbon associated with S_N2 remains unchanged after modifying the PES. Based on these calculations, we can conclude if there is no competing E2, the intrinsic reactivity of S_N2 can reach very high even though there exists steric congestion for the $F^- + (CH_3)_3CI$ reaction. In other words, the steric congestion cannot suppress the S_N2 reactivity for the $F^- + (CH_3)_3CI$ reaction (See details on Page 7 and Page 8 of the manuscript).

If the steric bulk is non-reactive group, but not the methyl group, the S_N2 reactivity can be very high. The conclusion is the nearly vanish of the S_N2 reactivity is not due to the steric congestion of the bulk, but the high reactivity of the bulk. This is the most important point we want to address in the current manuscript.

It is interesting to see more reactions, such as for other bases. We have to construct full-dimensional PESs and perform further dynamics simulations. Such work is ongoing in our group and will be published in near future.

Interestingly, the authors state in their Discussion section that the steric bulk makes the SN2 reaction more difficult to occur. This is in line with the current textbook explanation behind the retardation of the SN2 reactivity by bulky alkyl substituents and opposes the main conclusion of the authors.

Due to the above-mentioned inconsistencies in this work, I cannot recommend the

publication of this manuscript in Nature Communications.

There should be misunderstanding here. Based on those calculations, it is evident that the extremely small reactivity of S_N2 for the $F^-(CH_3)_3CI$ reaction is due to the inevitable competition with the highly reactive E2, but not the steric hindrance. We have changed the presentation in the Discussion section, to make things more clear. For example, the first paragraph of the Discussion section is rewritten as follows:

“Regarding the high reactivity of the E2 pathway, there are 9 hydrogen atoms in three methyl groups (here the bulk for the title reaction), either of which can react with the attacking F⁻. If the bulk consists of non-reactive groups, just as we want to demonstrate the similar way in this work by blocking the E2 pathway but maintaining the existence of bulk, the incoming F⁻ can attack the α -carbon easily and the S_N2 reactivity can reach very high. As a result, the nearly vanish of the S_N2 reactivity is not due to the steric congestion of the bulk, but the high reactivity of the bulk.”

The second paragraph, *“...it was found that the suppression of S_N2 for the prototype $F^- + {}^1C_4H_9I$ reaction results from the high reactivity of the three methyl groups, i.e., E2 pathway. With the aid of current analytical PES, we verify that in the absence of E2 competition, the intrinsic reactivity of S_N2 can reach that high even though there exists steric congestion.”*

Reviewer #3 (Remarks to the Author):

In this article, the authors report an unprecedented full-dimensional (39-dimensional) machine learning-based potential energy surface for the 15-atom $F^- + (CH_3)_3CI$ reaction, and the QCT simulation is carried out to study the elimination (E2) and bimolecular nucleophilic substitution (S_N2). I have the following comments for this article.

1. This is a multi-dimensional potential (39-dimensional), but the authors only show a schematic plot of the PES, without a contour plot of the important coordinates of the reaction, one could not see the relative positions of transition states (TS). In the schematic PES, all the TS are in the same position, which is not possible. Maybe the findings of the study also depend on the positions of the TS. Investigation should be done on this.

We have shown the contour plots of four pathways, i.e., back-side attack S_N2 , front-side attack S_N2 , anti-E2 and syn-E2 pathways, and added the following paragraph on Page 6. *“To further demonstrate the accurate behavior of full-dimensional PES, we show two contour plots of back-side attack S_N2 and anti-E2 pathways in Fig. 2. The contours of front-side attack S_N2 and syn-E2 pathways with higher energy profiles are depicted in Fig. S3 and Fig. S4, respectively. These contours are plotted using the full-dimensional optimization on the global*

full-dimensional PES. As seen, these contours are quite smooth, and the position and energies of reactants, pre-reaction minimums, transition states, post-reaction minimums and products, are all well described. To the best of our knowledge, this is also the first time these contours can be made for such large reactive systems. These contours further reflect the high accuracy of the current fitting and reliability of 15-atom multi-channel PES as well as the strong capability of the FI-NN approach.”

2.The authors state a full dimensional machine-learning PES, why does this reaction only has E2 and SN2 mechanisms on the full-d PES? I am not convinced there are only two mechanisms presented for this system. The front-side attack ? the double-inversion ? Other mechanisms might occur but have the same product as the two mechanisms , which leading the suppress one and enhance the other of the two mechanisms here.

The reaction only has E2 and S_N2 pathways (product channels) on the full-dimensional PES. This is exactly what we get by running trajectories on the full-dimensional PES. The referee mentioned the front-side attack and double-inversion mechanisms. Please note that front-side attack, double-inversion, or back-side attack or other indirect mechanisms are all mechanisms in the S_N2 reaction pathway. For the current F⁻+(CH₃)₃CI reaction, the dynamics calculations show negligible S_N2 reactivity (see blue curve in Fig. 5). We did not find front-side attack or double-inversion mechanism of S_N2, but limited back-side attack trajectories of S_N2. On page 6, we have added the following statements. *“In addition, the limited S_N2 reactivity results from exclusively the back-side attack mechanism, and no front-side attack mechanism was seen presumably due to the much higher barrier. Thus, “S_N2” below refers to back-side attack S_N2 for simplicity.”* For E2, we calculated the branching ratio of anti-E2 and syn-E2 pathways in Fig. S5, and also direct and indirect mechanism of E2 in Fig. S6.

On the modified PES, the S_N2 reactivity increases rapidly (Fig. 5). We found the S_N2 reaction on the modified PES proceeds via the direct or indirect back-side attack mechanism. The mechanism of direct back-side attack rises with the increase of collision energy, as indicated on Page 7.

3.The authors also add a repulsive potential between F-H in the vicinity of the E2 transition state of the global PES. The Equ (2) is a one dimensional, however the reaction is 39 dimensional with all the coordinates coupled, how the authors add this potential without change the behavior of other coordinates?

Without a through study of the possible mechanisms on the PES, I am not convinced the authors' argument is strong enough to support the current conclusion.

The referee seems not convinced by the IRCs on the original PES and modified

PES. In the revised manuscript, we have shown the contour plots of the S_N2 reaction on the original full-dimensional PES (Fig. 2a), and also on the modified PES (Fig. S7). We can see exactly the same results between the two full-dimensional PESs.

In addition, we have randomly selected two S_N2 trajectories and calculated the potential energies of configurations along the two trajectories. As shown in Fig. S8, the energies from the original and modified PESs are basically the same, and they are all well reproduced by CAM-XYG3/AVTZ(-PP) calculations. As a result, the inclusion of repulsive potential in Eq. (2) does not change the full-dimensional PES of S_N2 or the behavior of all 39 degrees of freedom of S_N2 , but completely blocks the E2 pathway.

Reviewers' comments:

Reviewer #1 (Remarks to the Author):

In my opinion, the authors have well addresses my comments. I have also read the reviews of the other two Referees, and I have to say that I mostly disagree with their comments and their recommendations and I agree with the authors' choice of appealing the editorial decision. I think the authors have properly addressed the other two Reviewers' concerns and I agree with the authors' responses. I still strongly recommend this work for publication in Nature Communications, because the applied methodology goes beyond state-of-the-art from some aspects and the new findings will move the field and science forward while attracting broad attention.

Specific comments:

I think Reviewer 2 does not realize that the present dynamics study on a full-dimensional potential energy surface goes beyond the old stationary-point-based or reduced-dimensional work. Thus, the present dynamics results provide the most reliable conclusion to date, which deserves publication even if it contradicts to conventional chemical thinking and/or previous literature based on less rigorous models.

In my view Reviewer 3's comments are minor and well addressed by the authors. I am confident to assess the comments of this Reviewer as my group has also studied front-side attack and double inversion, suggested by the Reviewer.

I have found a typo in the SI, "10.0 Bobr", but this can be corrected at the proof stage.

Reviewer #2 (Remarks to the Author):

The authors of the manuscript entitled "Unexpected Steric Hindrance Failure in de F⁻ + (CH₃)CI SN₂ Reaction" have, in their rebuttal, not convinced this reviewer to suggest publication of this work in Nature Communications. This work is lacking convincing evidence for the bold claim that the competing E₂ pathway, instead of steric hindrance, determines the small reactivity of SN₂ for the title reaction (see below).

Fu, Zhang, and co-workers base their conclusion solely on the investigation of one single reaction(!), namely, $F^- + (CH_3)Cl$. This reaction is an idealized case of the $S_N2/E2$ competition because F^- is an extremely potent Lewis base and hence, by definition, prefers to follow the $E2$ pathway, even for the $S_N2/E2$ competition with the simple and almost not sterically encumbered substrate haloethane, as mentioned by the authors (Page 2; Refs 24-26). Thus, the selected reaction intrinsically goes via an $E2$ pathway, and hence increasing the steric bulk around the alpha-carbon will not greatly affect the preference in this competition reaction. One should, therefore, study reactions that involve weaker nucleophiles, such as Cl^- or Br^- , and have an intrinsic preference for the S_N2 pathway, see Refs. 24-26. For these types of nucleophile, increasing the steric bulk on the substrate might induce a switch in preferred pathway from S_N2 to $E2$.

Nevertheless, in order to accurately comment on the, according to the authors, non-existing effect of increasing the steric bulk around the alpha-carbon on the S_N2 reactivity, one should compare the reactivity of substrates when systematically increasing steric bulk around the alpha-carbon, that is, MeI, EtI, *i*PrI, and *t*Bul. Only in this way, one can assess the effect of increasing the steric bulk on the S_N2 reactivity and hence can one comment on why the S_N2 reactivity becomes suppressed when the substrate becomes more sterically encumbered.

The point-by-point response to the comments are presented as follows. All the referees' comments are reproduced before the responses (indicated by blue fonts).

Reviewers' comments:

Reviewer #1 (Remarks to the Author):

In my opinion, the authors have well addressed my comments. I have also read the reviews of the other two Referees, and I have to say that I mostly disagree with their comments and their recommendations and I agree with the authors' choice of appealing the editorial decision. I think the authors have properly addressed the other two Reviewers' concerns and I agree with the authors' responses. I still strongly recommend this work for publication in Nature Communications, because the applied methodology goes beyond state-of-the-art from some aspects and the new findings will move the field and science forward while attracting broad attention.

Specific comments:

I think Reviewer 2 does not realize that the present dynamics study on a full-dimensional potential energy surface goes beyond the old stationary-point-based or reduced-dimensional work. Thus, the present dynamics results provide the most reliable conclusion to date, which deserves publication even if it contradicts to conventional chemical thinking and/or previous literature based on less rigorous models.

In my view Reviewer 3's comments are minor and well addressed by the authors. I am confident to assess the comments of this Reviewer as my group has also studied front-side attack and double inversion, suggested by the Reviewer.

I have found a typo in the SI, "10.0 Bobr", but this can be corrected at the proof stage.

We thank the Referee for positive comments.

Reviewer #2 (Remarks to the Author):

Fu, Zhang, and co-workers base their conclusion solely on the investigation of one single reaction(!), namely, $F^- + (CH_3)_3CI$.

Strange! Our conclusion is indeed for this single reaction, as indicated in the title "Unexpected steric hindrance failure in the \$F^- + (CH_3)_3CI\$ SN2 reaction" and the whole manuscript. We never claimed the conclusion is for all reactions.

This reaction is an idealized case of the SN2/E2 competition because F⁻ is an extremely potent Lewis base and hence, by definition, prefers to follow the E2 pathway, even for the SN2/E2 competition with the simple and almost not sterically encumbered substrate haloethane, as mentioned by the authors (Page 2; Refs 24-26). Thus, the selected reaction intrinsically goes via an E2 pathway, and hence increasing the steric bulk around the alpha-carbon will not greatly affect the preference in this competition reaction. One should, therefore, study reactions that involve weaker nucleophiles, such as Cl⁻ or Br⁻, and have an intrinsic preference for the SN2 pathway, see Refs. 24-26. For these types of nucleophile, increasing the steric bulk on the substrate might induce a switch in preferred pathway from SN2 to E2.

The above comments are totally based on the stationary-point-based calculations. Such calculations including Refs. 24-26 give energies of transition states for a series of reactions, which suggest the preference of reactivity for E2 or SN2 based on the barrier heights. As shown in Fig. 1 of our manuscript, both SN2 and anti-E2 are reactions with submerged barriers, -0.58 eV vs -0.82 eV, and the former is more exoergic. From this schematic of PES, one can only have a rough idea that the reactivity of E2 will be larger than SN2, but I don't think it is possible to infer that the SN2 reactivity is negligible compared with E2, as observed experimentally. Indeed, our calculation showed that the **SN2 reactivity nearly vanishes**, but what is the true origin of **nearly vanish of SN2** reactivity? Is the intrinsic reactivity of SN2 negligible as compared with E2 due to the small difference on the negative barrier height?

Interestingly, we obtained the intrinsic reactivity of SN2 by blocking the E2 pathway. This cannot be accomplished by any stationary-point-based calculations or experiment. We find the SN2 reactivity can reach very high when the E2 channel was blocked. This means the intrinsic reactivity of SN2 for F⁻ + (CH₃)₃CI is actually large. The nearly vanish of the S_N2 reactivity for F⁻ + (CH₃)₃CI is not due to the steric congestion of the bulk, but the high reactivity of the bulk which consumes all incoming F⁻. In other word, if the E2 reactivity was not that high to consume all incoming F⁻, one would observe the SN2 reactivity comparable to E2. Therefore, our finding is different from what Referee 2 thinks based on the previous stationary-point-based calculations.

Nevertheless, in order to accurately comment on the, according to the authors, non-existing effect of increasing the steric bulk around the alpha-carbon on the SN2 reactivity, one should compare the reactivity of substrates when systematically

increasing steric bulk around the alpha-carbon, that is, MeI, EtI, iPrI, and tBuI. Only in this way, one can assess the effect of increasing the steric bulk on the SN2 reactivity and hence can one comment on why the SN2 reactivity becomes suppressed when the substrate becomes more sterically encumbered.

We will construct full-dimensional PESs and perform dynamics calculations for a series of reactions in future. Those results would not change the current conclusion for $F^- + (CH_3)_3CI$. Probably Referee 2 does not know how many computational costs and efforts would be devoted to one single reaction using our rigorous approach.

REVIEWER COMMENTS

Reviewer #4 (Remarks to the Author):

I have no issue with the approach of the study, and the amount of sampling is particularly impressive. The idea of artificially blocking the E2 pathway via alternating the PES is interesting too. The agreement between the computation and experiment (time of flight and scattering angle distribution) is remarkable.

That being said, I wish the authors had chosen to study at least one more reaction system to make their claim stronger. I do agree with reviewer #2 that F⁻ is an extremely potent Lewis base and the reaction involving Cl⁻ or Br⁻ would be a more appropriate system to study the competition between SN2 and E2. What reviewer #2 argues is indeed the current understanding and should be held as so until proven otherwise by experiments and dynamics simulations. I also would like to suggest that if the authors decide to study Cl⁻ or Br⁻, 4.2 million trajectories seem unnecessarily too many.

I disagree with other systems (e.g., MeI, EtI, iPrI, and tBuI) that reviewer #2 suggested, as I think they are too excessive for one single publication.

Reviewer #5 (Remarks to the Author):

The authors developed a 39-dimensional potential energy surface (PES) for F⁻ + (CH₃)₃CI reaction by fundamental invariant neural network approach. They performed extensive quasiclassical trajectory simulations on the PES. The integral cross sections (ICS) show that E2 pathway suppresses SN2 pathway by a factor of over 50. To reveal the intrinsic reactivity of SN2 pathway for F⁻ + (CH₃)₃CI reaction, the authors modified the PES by blocking the E2 path via adding a repulsive potential between F⁻ and beta-H. QCT simulations on the modified PES shows the ICS of SN2 pathway increases and becomes comparable to the ICS of E2 path on original PES. They conclude that the intrinsic reactivity of SN2 pathway is actually quite high, "if the E2 reactivity was not that high to consume all incoming F⁻, one would observe the SN2 reactivity comparable to E2". So far, everything is correct. However, the authors went too far to state this is a "steric hindrance failure" and to state "As opposed to the textbook example of the steric retardation of SN2 reactivity by bulky substitution...". Therefore, the current form of the manuscript cannot be accepted to the journal of Nature Communications. I think further revision is needed before it to be reconsidered.

On the one hand, I agree with Reviewer #1 that “the applied methodology goes beyond state-of-the-art from some aspects”. Undoubtedly, developing a 39-dimensional is a breakthrough and blocking the E2 pathway to reveal the intrinsic SN2 reactivity of titled reaction is an approach that cannot be accomplished by experiments and can be used in future studies. Nevertheless, it is important to think within the framework of common understanding of “steric hindrance”.

In my opinion, the main issue of the article is that the observation of “the SN2 reactivity is comparable to E2” when E2 is blocked doesn’t lead to the conclusion of “steric hindrance failure”. This is the main concern of the second reviewer and myself.

In organic chemistry, steric hindrance refers to the phenomenon that reactivity decreases as the substrate becomes bulkier. I agree with second reviewer that to prove “steric hindrance fails”, one should show the reactivity doesn’t decrease as the substrate becomes bulkier systematically. Ideally, one should compare the “intrinsic” SN2 reactivity of CH3I, C2H5I, C3H7I and (CH3)3CI, or at least compare the reactivity of CH3I and (CH3)3CI. With available data and literature, I think the authors can consider following comparisons.

1. How are the cross sections of (CH3)3CI (this work) compare to that of F- + CH3I? Gabor group constructed a PES for F- + CH3I reaction (Chemical Science 2017, 8 (4), 3164-3170) and reported cross sections at collision energies of 1.0 to 50 kcal/mol. These values can be compared with the values of F- + (CH3)3CI on modified PES.

2. How are the E2 reactivity change as the substrate becomes bulkier? If existing experiments on analogue systems show the E2 reactivity doesn’t decrease as the substrate becomes bulkier, then when the intrinsic SN2 reactivity of (CH3)3CI becomes comparable to that of E2, it should be comparable to the intrinsic reactivity of CH3I, C2H5I, C3H7I.

3. How are the overall barrier and internal barrier changes from CH3I to (CH3)3CI as they react with F-?

If “steric hindrance failure” is proved for the titled system, the authors shall explain why and why it is “opposed to the textbook example”. What present in textbook is reactions in liquid phase and their transition states are usually higher than reactants. In this work, the reaction is in gas-phase, the transition states are much lower than reactants, i.e. submerged barrier. The reason why the reactions proceed to SN2 path when E2 is blocked is because the prereaction complex well is deep, thus crossing the SN2 transition states to form SN2 products is much easier than dissociates back to reactants, especially at low collision energy. If the barrier is higher and the pre-reaction complex well is not as deep, say changing the nucleophile to Cl-/Br-/I-, the scenario would change. The authors shall emphasis the reaction is gas-phase reaction in the title.

Some minor issues.

No abbreviation is provided for QCT in the manuscript.

Page 11, at the end of first paragraph, the N of SN2 should be subscript.

We have made considerable efforts to improve our manuscript and have addressed all the points raised by the referees. The point-by-point response to the comments are presented as follows. All the referees' comments are reproduced before the responses (indicated by blue fonts).

Reviewer #4 (Remarks to the Author):

I have no issue with the approach of the study, and the amount of sampling is particularly impressive. The idea of artificially blocking the E2 pathway via alternating the PES is interesting too. The agreement between the computation and experiment (time of flight and scattering angle distribution) is remarkable.

That being said, I wish the authors had chosen to study at least one more reaction system to make their claim stronger. I do agree with reviewer #2 that F⁻ is an extremely potent Lewis base and the reaction involving Cl⁻ or Br⁻ would be a more appropriate system to study the competition between S_N2 and E2. What reviewer #2 argues is indeed the current understanding and should be held as so until proven otherwise by experiments and dynamics simulations. I also would like to suggest that if the authors decide to study Cl⁻ or Br⁻, 4.2 million trajectories seem unnecessarily too many.

I disagree with other systems (e.g., MeI, EtI, iPrI, and tBuI) that reviewer #2 suggested, as I think they are too excessive for one single publication.

We thank Reviewer #4 for positive comments on our work. As suggested by this reviewer, maybe one more reaction involving Cl⁻ or Br⁻ should be studied to make our claim stronger. Therefore, we chose to study one more reaction of similar type involving Cl⁻, namely, Cl⁻+C(CH₃)₃I. A full-dimensional potential energy surface (PES) for Cl⁻+C(CH₃)₃I has been developed using the same approach for F⁻+C(CH₃)₃I. The QCT calculations were performed based on this new PES to investigate the competition between S_N2 and E2 and also steric effects. The following details and Fig. S10 and Fig. S11 have been included in Sec. III (PES and dynamics of Cl⁻+C(CH₃)₃I) in the supplementary information. "In addition to the title F⁻ + (CH₃)₃CI reaction, we investigated one more reaction, Cl⁻ + (CH₃)₃CI, to make our claim stronger. The theoretical and computational approaches used for Cl⁻ + (CH₃)₃CI are basically the same as those for F⁻ + (CH₃)₃CI, and thus a brief description is given here. A full-dimensional PES of Cl⁻ + (CH₃)₃CI was developed by FI-NN fitting to about 270 000 data points. These data points were calculated with the ab initio level of CAM-

XYG3/aug-cc-pVTZ(aug-cc-pVTZ-PP for iodine atom). The overall RMSE of the fitted PES is 18.7 meV. Figure S10 shows the schematic of the PES, with important stationary-point geometries and energies illustrated. We can see a different minimum energy path of $\text{Cl}^- + (\text{CH}_3)_3\text{CI}$, as compared with that of $\text{F}^- + (\text{CH}_3)_3\text{CI}$. For instance, the barriers are much higher, the pre-reaction and post-reaction wells are shallower, and the products are not that exothermic for the Cl^- involved E2 and $\text{S}_{\text{N}}2$ reactions. Specifically, the $\text{S}_{\text{N}}2$ pathway has an overall barrier of 0.23 eV, which is slightly higher than the barrier of 0.17 eV for the E2 pathway. The $\text{S}_{\text{N}}2$ reaction shares an ion-dipole bound pre-reaction well with the E2 pathway, which lies -0.74 eV below the reactant asymptote. The $\text{S}_{\text{N}}2$ reaction is exothermic, but the E2 pathway is endothermic with the product energy 0.31 eV above the reactant energy.

The new PES was also modified by adding a similar repulsive potential as has been used for the F^- reaction, which blocks the E2 pathway completely but nothing is changed for $\text{S}_{\text{N}}2$. Standard QCT calculations for $\text{Cl}^- + (\text{CH}_3)_3\text{CI}$ were carried out on the new PES and also on the modified PES at collision energies ranging from 0.4 eV to 1.1 eV, for the $(\text{CH}_3)_3\text{CI}$ reactant initially in the ground rovibrational state. A total of roughly 1.3 million trajectories were run. The computed integral cross sections for the E2 and $\text{S}_{\text{N}}2$ pathways on the new PES as well as on the modified PES are shown in Fig. S11. The $\text{S}_{\text{N}}2$ cross section on the new PES is only 4 times smaller than that of E2, indicating the reactivities of the two pathways are in the same order of magnitude. This is understandable because the $\text{S}_{\text{N}}2$ pathway has a higher barrier, although it is more exothermic. After the E2 pathway was completely blocked on the modified PES, the $\text{S}_{\text{N}}2$ reactivity rises remarkably by about a factor of 2, which is about one half of the E2 reactivity. Hence the intrinsic $\text{S}_{\text{N}}2$ reactivity is closely related to the competition of E2 for $\text{Cl}^- + (\text{CH}_3)_3\text{CI}$, but not the steric hindrance, as we concluded for $\text{F}^- + (\text{CH}_3)_3\text{CI}$.

For the $\text{F}^- + (\text{CH}_3)_3\text{CI}$ reaction, the transition states are much lower than reactants, i.e., submerged barrier. The reason why the reactions almost proceed to $\text{S}_{\text{N}}2$ when E2 is blocked is because the pre-reaction complex well is deep, thus crossing the $\text{S}_{\text{N}}2$ transition state to form $\text{S}_{\text{N}}2$ products is much easier than dissociating back to reactants, especially at low collision energies. For $\text{Cl}^- + (\text{CH}_3)_3\text{CI}$, the transition state of $\text{S}_{\text{N}}2$ is higher than reactants and the pre-reaction complex well is not as deep, hence some fractions of reaction would dissociate back to reactants and some would proceed to $\text{S}_{\text{N}}2$ when E2 is blocked.”

Reviewer #5 (Remarks to the Author):

The authors developed a 39-dimensional potential energy surface (PES) for $F^- + (CH_3)_3CI$ reaction by fundamental invariant neural network approach. They performed extensive quasiclassical trajectory simulations on the PES. The integral cross sections (ICS) show that E2 pathway suppresses SN2 pathway by a factor of over 50. To reveal the intrinsic reactivity of SN2 pathway for $F^- + (CH_3)_3CI$ reaction, the authors modified the PES by blocking the E2 path via adding a repulsive potential between F- and beta-H. QCT simulations on the modified PES shows the ICS of SN2 pathway increases and becomes comparable to the ICS of E2 path on original PES. They conclude that the intrinsic reactivity of SN2 pathway is actually quite high, “if the E2 reactivity was not that high to consume all incoming F-, one would observe the SN2 reactivity comparable to E2”. So far, everything is correct. However, the authors went too far to state this is a “steric hindrance failure” and to state “As opposed to the textbook example of the steric retardation of SN2 reactivity by bulky substitution...”. Therefore, the current form of the manuscript cannot be accepted to the journal of Nature Communications. I think further revision is needed before it to be reconsidered.

On the one hand, I agree with Reviewer #1 that “the applied methodology goes beyond state-of-the-art from some aspects”. Undoubtedly, developing a 39-dimensional is a breakthrough and blocking the E2 pathway to reveal the intrinsic SN2 reactivity of titled reaction is an approach that cannot be accomplished by experiments and can be used in future studies. Nevertheless, it is important to think within the framework of common understanding of “steric hindrance”.

In my opinion, the main issue of the article is that the observation of “the SN2 reactivity is comparable to E2” when E2 is blocked doesn’t lead to the conclusion of “steric hindrance failure”. This is the main concern of the second reviewer and myself.

In organic chemistry, steric hindrance refers to the phenomenon that reactivity decreases as the substrate becomes bulkier. I agree with second reviewer that to prove “steric hindrance fails”, one should show the reactivity doesn’t decrease as the substrate becomes bulkier systematically. Ideally, one should compare the “intrinsic” SN2 reactivity of CH_3I , C_2H_5I , C_3H_7I and $(CH_3)_3CI$, or at least compare the reactivity of CH_3I and $(CH_3)_3CI$. With available data and literature, I think the authors can consider following comparisons.

1. How are the cross sections of $(\text{CH}_3)_3\text{CI}$ (this work) compare to that of $\text{F}^- + \text{CH}_3\text{I}$? Gabor group constructed a PES for $\text{F}^- + \text{CH}_3\text{I}$ reaction (Chemical Science 2017, 8 (4), 3164-3170) and reported cross sections at collision energies of 1.0 to 50 kcal/mol. These values can be compared with the values of $\text{F}^- + (\text{CH}_3)_3\text{CI}$ on modified PES.

We have compared the cross sections of $\text{F}^- + (\text{CH}_3)_3\text{CI}$ on the modified PES with those results of $\text{F}^- + \text{CH}_3\text{I}$ on the PES developed by Gabor group and also on the new FI-NN PES developed by us in Fig. S9. The following paragraph has been included on Page 11 of the manuscript. “It is also important to compare the intrinsic $\text{S}_{\text{N}}2$ reactivity of $\text{F}^- + (\text{CH}_3)_3\text{CI}$ with that of $\text{F}^- + \text{CH}_3\text{I}$. Figure S9 shows the cross sections of $\text{F}^- + (\text{CH}_3)_3\text{CI}$ on the modified PES and $\text{F}^- + \text{CH}_3\text{I}$ on the PES developed by Czako et al[5] as well as on the PES developed by us. The cross sections of $\text{F}^- + \text{CH}_3\text{I}$ on the two different PESs are consistent, indicating both the two PESs are sufficiently accurate. It is interesting that the intrinsic $\text{S}_{\text{N}}2$ reactivity of $(\text{CH}_3)_3\text{CI}$ is even larger than that of CH_3I , probably due to more exothermic energy of $(\text{CH}_3)_3\text{CI}$. These results unambiguously show the $\text{S}_{\text{N}}2$ reactivity doesn't decrease as the substrate becomes bulkier, which is also the evidence of steric hindrance failure.”

2. How are the E2 reactivity change as the substrate becomes bulkier? If existing experiments on analogue systems show the E2 reactivity doesn't decrease as the substrate becomes bulkier, then when the intrinsic $\text{S}_{\text{N}}2$ reactivity of $(\text{CH}_3)_3\text{CI}$ becomes comparable to that of E2, it should be comparable to the intrinsic reactivity of CH_3I , $\text{C}_2\text{H}_5\text{I}$, $\text{C}_3\text{H}_7\text{I}$.

To the best of our knowledge, no existing experiments show the variation of E2 reactivity when the substrate becomes bulkier. However, we have shown that the intrinsic $\text{S}_{\text{N}}2$ reactivity of $(\text{CH}_3)_3\text{CI}$ becomes comparable to that of E2, and it is also comparable to the intrinsic reactivity of CH_3I . These comments were also addressed in Question 1.

3. How are the overall barrier and internal barrier changes from CH_3I to $(\text{CH}_3)_3\text{CI}$ as they react with F^- ?

The following table shows the changes of overall barrier and internal barrier for the F⁻ reaction from CH₃I to (CH₃)₃CI. The results for F⁻+ⁱPrI are not included because they were not reported in the literature. However, we can see obvious trend as the substrate becomes bulkier. The overall barrier and internal barrier of S_N2 increase when the substrate becomes bulkier, but not dramatically. We claim that the intrinsic reactivity can only be determined by full-dimensional dynamics calculations based on a full-dimensional PES, but not the stationary-point characterization.

	Overall barrier (eV)	Internal barrier (eV)
F ⁻ +MeI ^a	-0.75	0.007
F ⁻ +EtI ^b	-0.73	0.12
F ⁻ + ⁱ PrI	/	/
F ⁻ + ^t BuI ^c	-0.53	0.58

^aCCSD(T)-F12b/AVTZ-PP by Czakó *et al*, *Chem. Sci.*, 2017, 8, 3164.

^bCCSD(T)PP/t//MP2/ECP/d by Hase *et al*, *J. Phys. Chem A*, 2017, 121, 1078.

^cCCSD(T)-F12b/AVTZ-PP, present work.

If “steric hindrance failure” is proved for the titled system, the authors shall explain why and why it is “opposed to the textbook example”. What present in textbook is reactions in liquid phase and their transition states are usually higher than reactants. In this work, the reaction is in gas-phase, the transition states are much lower than reactants, i.e. submerged barrier. The reason why the reactions proceed to S_N2 path when E2 is blocked is because the pre-reaction complex well is deep, thus crossing the S_N2 transition states to form S_N2 products is much easier than dissociates back to reactants, especially at low collision energy. If the barrier is higher and the pre-reaction complex well is not as deep, say changing the nucleophile to Cl⁻/Br⁻/I⁻, the scenario would change. The authors shall emphasis the reaction is gas-phase reaction in the title.

We have added “gas phase” in the title, and thus “gas phase” was emphasized. In Discussion Section, we have added the useful comments of this point as follows. “

For the F⁻ + (CH₃)₃CI gas phase reaction, the transition states are much lower than reactants, i.e. submerged barrier. The reason why the reactions almost proceed to S_N2 when E2 is blocked is because the pre-reaction complex well is deep, thus crossing the S_N2 transition state to form S_N2 products is much easier than dissociating back to reactants, especially at low collision energies. For the gas phase Cl⁻ + (CH₃)₃CI, the

transition state of S_N2 is about 0.23 eV higher than reactants and the pre-reaction complex well is not as deep, hence some fractions of reaction would dissociate back to reactants and some would proceed to S_N2 when E2 is blocked. More details of $Cl^- + (CH_3)_3CI$ were given in the supplementary information. We expect the similar scenario in textbook liquid phase reactions, whose transition states are much higher than reactants.”

Some minor issues.

No abbreviation is provided for QCT in the manuscript.

Fixed.

Page 11, at the end of first paragraph, the N of S_N2 should be subscript.

Fixed.

NCOMMS-21-36391C

The authors have addressed my previous concerns, that the conclusion of E2 vs. S_N2 is too broad with just one reaction (e.g., $F^- + (CH_3)_3Cl$). The authors changed the halogen anion and included a new $Cl^- + (CH_3)_3Cl$ reaction (with the same approach and analysis) and found that the conclusion is largely unchanged. As indicated in the previous round of review, I have no problem with the approach (e.g., fit a hyper dimension potential energy surface to run dynamics that approach the level of accuracy of AIMD with DFT), which shows good agreement with the experiment. However, it should be pointed out that the analysis of the dynamics of the $Cl^- + (CH_3)_3Cl$ reaction is entirely in the SI, with just one pointer sentences which reads as “More details of $Cl^- + (CH_3)_3Cl$ were given in the supplementary information” in the main manuscript. I am troubled by this arrangement, as comparing to another reaction is a key component, instead of an afterthought, for the authors to make such claim. Therefore, I urge the authors to move the analysis of the $Cl^- + (CH_3)_3Cl$ reaction to the main text.

There are two other relatively minor points should be addressed in the published form of the manuscript.

1. On page 8, the authors claimed that the time of flight (ToF) of $F^- + (CH_3)_3Cl$ becomes independent to collision energy and claimed that i. the reaction goes through indirect dynamics (i.e., forming intermediates) and ii. the product becomes more rovibrationally excited as the increase of collision energy. This statement is intuitive, but evidence (e.g., i. % of indirect reactive trajectories and ii. average internal energy of the product) should be added to the SI.
2. On page 8, the author introduced a potential penalty to block the E2 reaction while claiming it does not impact the “intrinsic” S_N2 reaction. I found the idea of the “intrinsic” S_N2 is rather strange, because technically there is no such thing – S_N2 reaction is competing with other reactions (e.g., E2) so whatever measured from QCT should indeed be intrinsic. Therefore, I suggest the author to put the word “intrinsic” into quotations to highlight that it is an unusual definition. Further, I would suggest the authors to provide an updated version of the PES like Fig 1 with the potential penalty included in the SI.

Reviewer #5 (Remarks to the Author):

This is a modified manuscript concerning the steric hindrance of ion-molecule reactions that can proceed either SN2 or E2 pathways.

The authors made remarkable efforts and have answered all the issues raised by the referees, so I am glad to recommend the publication of this manuscript. One minor point for the statement " We expect the similar scenario in textbook liquid phase reactions, whose transition states are much higher than reactants." When solvent molecules are involved, the reaction will be affected by the solute-solvent interaction, so the scenario may be complicated.

The point-by-point response to the comments are presented as follows. All the referees' comments are reproduced before the responses (indicated by blue fonts).

Reviewer #4 (Remarks to the Author):

The authors have addressed my previous concerns, that the conclusion of E2 vs. SN2 is too broad with just one reaction (e.g., $F^- + (CH_3)_3CI$). The authors changed the halogen anion and included a new $Cl^- + (CH_3)_3CI$ reaction (with the same approach and analysis) and found that the conclusion is largely unchanged. As indicated in the previous round of review, I have no problem with the approach (e.g., fit a hyper dimension potential energy surface to run dynamics that approach the level of accuracy of AIMD with DFT), which shows good agreement with the experiment. However, it should be pointed out that the analysis of the dynamics of the $Cl^- + (CH_3)_3CI$ reaction is entirely in the SI, with just one pointer sentences which reads as "More details of $Cl^- + (CH_3)_3CI$ were given in the supplementary information" in the main manuscript. I am troubled by this arrangement, as comparing to another reaction is a key component, instead of an afterthought, for the authors to make such claim. Therefore, I urge the authors to move the analysis of the $Cl^- + (CH_3)_3CI$ reaction to the main text.

We have moved the section of $Cl^- + (CH_3)_3CI$ to Page 11 of the main text.

There are two other relatively minor points should be addressed in the published form of the manuscript.

1. On page 8, the authors claimed that the time of flight (ToF) of $F^- + (CH_3)_3CI$ becomes independent to collision energy and claimed that i. the reaction goes through indirect dynamics (i.e., forming intermediates) and ii. the product becomes more rovibrationally excited as the increase of collision energy. This statement is intuitive, but evidence (e.g., i. % of indirect reactive trajectories and ii. average internal energy of the product) should be added to the SI.

We did have a figure in SI (Fig. S5) for the cross sections of indirect and direct reactive trajectories. The internal energy distribution of product $(CH_3)_2CCH_2$ has been added in Fig. S7. We can see this distribution peaks at around 3.5 eV~4.0 eV, showing a tail up to 5.0~6.0 eV. The zero-point energy of $(CH_3)_2CCH_2$ is only 1.9 eV. Thus, the $(CH_3)_2CCH_2$ product is highly rovibrationally excited. In addition, the $(CH_3)_2CCH_2$ product becomes more rovibrationally excited as the increase of collision energy. We have included the above statements on Page 8 of the manuscript.

2. On page 8, the author introduced a potential penalty to block the E2 reaction while claiming it does not impact the “intrinsic” SN2 reaction. I found the idea of the “intrinsic” SN2 is rather strange, because technically there is no such thing – SN2 reaction is competing with other reactions (e.g., E2) so whatever measured from QCT should indeed be intrinsic. Therefore, I suggest the author to put the word “intrinsic” into quotations to highlight that it is an unusual definition. Further, I would suggest the authors to provide an updated version of the PES like Fig 1 with the potential penalty included in the SI.

We have put the word “intrinsic” into quotations in the manuscript. The PES of E2 with the SN2 pathway blocked is the same with the original PES in Fig. 1, despite of the exclusion of SN2.

Reviewer #5 (Remarks to the Author):

This is a modified manuscript concerning the steric hindrance of ion-molecule reactions that can proceed either SN2 or E2 pathways.

The authors made remarkable efforts and have answered all the issues raised by the referees, so I am glad to recommend the publication of this manuscript. One minor point for the statement " We expect the similar scenario in textbook liquid phase reactions, whose transition states are much higher than reactants." When solvent molecules are involved, the reaction will be affected by the solute-solvent interaction, so the scenario may be complicated.

We have changed this sentence into now read “We expect the similar scenario in other gas-phase SN2/E2 competitive reactions, whose transition states are much higher than reactants.”.